# Sulfonic Cryogels as Innovative Materials for Biotechnological Applications: Synthesis, Modification, and Biological Activity

**DOI:** 10.3390/ijms24032949

**Published:** 2023-02-02

**Authors:** Svetlana Laishevkina, Tatiana Kuleshova, Gayane Panova, Elena Ivan’kova, Olga Iakobson, Anatoly Dobrodumov, Natalia Shevchenko, Alexander Yakimansky

**Affiliations:** 1Institute of Macromolecular Compounds, Russian Academy of Sciences, Bolshoy pr., 31, 199004 Saint-Petersburg, Russia; 2Agrophysical Research Institute, Grazhdanskiy pr. 14, 195220 Saint-Petersburg, Russia

**Keywords:** sulfonic cryogel, swelling and deswelling, activated nanodiamonds, urea, seed germination

## Abstract

Polymeric hydrogels based on sulfo-containing comonomers are promising materials for biotechnological application, namely, for use as a system for delivering water and minerals during seed germination in conditions of an unstable moisture zone. In this work, cryogels based on 3-sulfopropyl methacrylate and 2-hydroxyethyl methacrylate copolymers were obtained by the cryotropic gelation method. The morphology, specific surface area, and swelling behaviors of cryogels are found to depend on the total concentration of monomers in the reaction system and the content of the gel fraction in cryogels. Cryogels formed in the presence of nanodiamonds are shown to exhibit high biological activity during the germination of *Lepidium sativum* L. variety Ajur seeds, which manifests itself by stimulating seed germination and a significant increase in the raw weight of sprouts. These results indicate that sulfonic cryogels have a high potential to improve seed germination and plant growth, proving that such cryogels can be used as environmentally friendly materials for agricultural applications.

## 1. Introduction

Hydrogels present three-dimensional structured “polymer–water” systems in which macromolecules are involved in a spatial network via relatively stable over time non-fluctuation bonds [1]. A hydrogel can be cross-linked through chemical or physical bonds (hydrogen bonds, ionic or hydrophobic interactions, as well as chain entanglement [2]). By their nature, hydrogels are natural or synthetic three-dimensional structures capable of equilibrium and reversible swelling in water and aqueous solutions. At the same time, depending on the degree of cross-linking and the nature of the polymer, the hydrogels are able to retain a large amount of water/solvent (hundreds and thousands of times more relative to their own weight [3]) while maintaining their shape. The hydrogel application scope is mostly determined by gel structure, the degree of “free swelling”, as well as its mechanical properties. Thus, due to their soft, porous structure resembling biological tissues, hydrogels have found wide application for the formation of materials for medical and biotechnological purposes (templates for growing cells, wound dressings, carriers for enzyme immobilization, systems for controlled release of drug substances) [4]. In addition, due to the hydrophilic surface, hydrogels are characterized by low values of free surface energy when interacting with biological fluids, which leads to very low adhesion of proteins and cells on their surface, allowing such hydrogels to be used as effective implants. Moreover, the softness and elasticity of hydrogels minimize mechanical irritation of surrounding tissues [5]. Flexibility and the possibility of modifying hydrogels with electrically conductive components make it possible to obtain materials promising for flexible bioelectronics and biosensorics [6,7,8]. Hydrogels capable of retaining a large amount of water are topical objects of practical interest due to the possibility of their use as effective sorbents in solving a number of environmental and agrotechnical problems [3]. 

Fresh water plays an important role in ensuring the growth of living organisms, including plants, and, taking into account that water reserves in the world are limited [9], there is a need to optimize the methods of using fresh water in the conditions of an unstable moisture zone. When considering the traditional way of growing plants, a significant part of the water and most of the fertilizer passes through the soil, “bypassing” the roots and mixing with groundwater. For such reasons, polymer hydrogels are increasingly used to ensure the growth and productivity of agricultural crops. Hydrogels in agriculture simultaneously perform several tasks. They are water reservoirs to reduce moisture loss in the soil [3,10,11] and conditioners, increasing the porosity of the soil and thus providing better oxygen circulation to the plant roots. Furthermore, depending on the chemical structure and the additives used, hydrogels can act as carriers of essential nutrients (micro- and macroelements) for their slow release during plant growth [10]. Researchers all over the world are trying to solve the problem of efficient use of fertilizers. It is known that after the introduction of carbamide fertilizers into the soil, they pass from a solid state to a gaseous state, as a result of which losses can reach 40–70% [12]. Moreover, carbamide fertilizers lead to the release of harmful NH_3_ and N_2_O, which contributes to global warming and increases environmental and climate problems [12,13,14]. To solve the problem of fast release of nitrogen from carbamide fertilizers, the use of hydrogel coatings or capsules is proposed [12]. At the same time, not coating the fertilizer with a polymer shell but instead introducing urea into the hydrogel structure is more efficient for the slow release of nitrogen. [10]. To sum up, the limitation of fresh water resources, inefficient use of fertilizers, and the need to protect the environment underlie a significant interest in the study of hydrogels, which are promising for agricultural use as a means of delivering water and nutrients to the root systems of plants.

The main requirements for hydrogels used for agricultural purposes are a high degree and rate of swelling and the ability to re-swell. [11]. These properties are mostly determined by the size of hydrogel pore [15]. Moreover, the degree of swelling depends on the nature of the polymer and the degree of cross-linking. Obviously, the lower density of cross-links, the higher degree of swelling [16]. At the same time, the influence of the chemical structure of hydrogels on the degree of swelling is a more complex dependence.

Nowadays, hydrogels based on natural polymers, such as starch [17,18,19], chitosan [20], or cellulose [3], are widely used in agriculture. Such hydrogels are biocompatible, biodegradable, and environmentally friendly. However, unmodified natural polymers are characterized by low swelling degree (about 100–200 g/g); in addition, their properties vary greatly depending on the type and place origin or production year, which strongly reduces the reproducibility of the results [3]. In this regard, many researchers either modify natural polymers with synthetic ones, or use only synthetic polymers, among which polyacrylic acid and its salts [10,21,22,23] or polyacrylamide [24] are most widely used. Such hydrogels are usually characterized by a degree of swelling in water of 300–500 g/g, which can be increased through various modifications by copolymerization, as well as the introduction of various additives (dopants).

Increasing the proportion of ionic groups (such as COOH [10] or SO_3_H [23,25,26,27,28]) in the hydrogel structure leads to the maximization of gel swelling degree. Since the sulfo group exhibits stronger acidic properties, the synthesis of hydrogels based on sulfo-containing comonomers is a promising means of solving the problem of obtaining biotechnological materials. Zaborina et al. [28] have shown that the copolymerization of potassium 3-sulfopropyl methacrylate and *N*,*N*-dimethylacrylamide makes it possible to obtain a hydrogel with the degree of swelling equal to 1850 g/g, while the degree of swelling of the sodium polyacrylate gel was 1700 g/g. Similar results are observed for other acrylate sulfonic hydrogels. Scognamillo et al. [26] have synthesized hydrogels based on 3-sulfopropylacryl and N-isopropylacrylamide. The swelling degree of such hydrogels raised linearly with the increase in 3-sulfopropylacrylate concentration and the cross-linking agent (*N*,*N*′-methylene-bis-acrylamide (MBA)) content decrease. Guan et al. [23] have prepared hydrogels based on acrylamide and 2-acrylamido-2-methyl-1-propanesulfonic acid, the degree of swelling of which exceeded 1000 g/g.

The introduction of various natural or synthetic additives into the structure of hydrogels contributes to an increase in their degree of swelling. The sodium humate [23], maize bran [29], laterite [30], graphene oxide [31,32], etc., are discussed in the literature as hydrogel dopants. The latter dopant is shown to be an effective water-retaining agent for the soil, while being non-toxic and having a positive effect on plant photosynthesis [31]. Despite this, a limited number of works where graphene oxide is used as one of the components of hydrogel for agriculture application can be found [31,32]. At the same time, detonation nanodiamonds (DNDs) that are similar to graphene oxide in structure and are also good moisture absorbers and antibacterial agents, such as graphene oxide [33], are practically not studied [34]. DNDs are of particular interest because they contain a large number of hydroxyl, epoxy, carbonyl, and carboxyl groups, resulting in high hydrophilicity. [32]. The introduction of DNDs into the hydrogel structure can not only improve its degree of swelling, but also prevent the growth of pathogenic bacteria in the soil.

Despite the large number of works devoted to the preparation of polymer hydrogels, the problem of developing a hydrogel material that satisfies a wide range of requirements for possible biotechnological purposes (degree of swelling, toxicity, porosity, etc.), and at the same time is characterized by the ease of access and relatively simple production technology, still remains unresolved.

Thus, the tasks of this work are the following: (1) the formation of porous cryogels based on sulfo-containing comonomer (as 3-sulfopropyl methacrylate) capable of retaining water at a concentration of at least 500 g/g; (2) the investigation of the effect of the composition of the reaction mixture (concentration and ratio of monomers, additives of various natures (starch, hydroxypropyl cellulose, urea, DNDs)) on the degree of swelling of the resulting cryogels; (3) the study of the structure and properties of cryogels by SEM, ATR-FTIR-, and ^13^C NMR spectroscopy; (4) the elucidation of the potential of the obtained cryogel for use as a biotechnological material for the germination of seeds (on the example of *Lepidium sativum* L. variety Ajur).

## 2. Results and Discussion

### 2.1. Synthesis of Cryogels

Cryogels were synthesized via cryotropic gelation during the copolymerization of potassium 3-sulfopropyl methacrylate (SPM), which serves as a source of SO_3_^−^ groups, and hydrophilic 2-hydroxyethyl methacrylate (HEMA) containing OH groups in the structure. Synthesis conditions and results (average of at least three experiments) are shown in Table 1. The reaction was carried out at −18 °C. Cryotropic gelation is chosen as a synthesis method due to the possibility of forming gels with a highly porous structure [28,35]. It is known that, the pore space is first filled with a capillary solvent (~90–95% of the total amount of solvent) for highly porous gels, and then the polymer wall gradually swells with a solvate-bound solvent. [25] As a result, such hydrogels are characterized by the maximum values of the swelling degree. It should be noted that during cryotropic gelation, the reaction proceeds in a non-freezing microphase around ice crystals (Figure 1), which makes it possible to control the size and structure of pores by changing the freezing temperature of the reaction system [27]. Such gels are characterized by explosive/rapid swelling behavior [27], and it is possible to reduce the concentration of monomers used for the synthesis due to the effect of concentration of the reaction system, which is an important from an economic point of view. For example, the total monomer concentration (SPM&HEMA) equal to 1 mol/L is not sufficient for the hydrogel formation at a temperature of 25 °C (Table 1, experiment 1). While hydrogel is obtained both at a concentration of 1 mol/L and at a lower concentration of 0.5 mol/L when the process is carried out at −18 °C (Table 1, experiments 2 and 3, respectively). 

Since no cross-linking agent is added into the system under study, the formation of a three-dimensional polymer network is ensured by non-covalent interactions, namely, by hydrogen bonds between the hydrogen of the HEMA hydroxyl group and the oxygen of the SPM sulfonate group. Increasing the monomer concentration from 0.5 to 5 mol/L results in a decrease in the hydrogel swelling degree (Figure 2a, blue line). The hydrogel swelling degree is known to demonstrate an opposite dependence on the cross-linking density. In our case, an increase in concentration of comonomers, even up to 1 mol/L, leads to a sharp increase in the density of hydrogen bonds in the structure of hydrogels, which results to reduce the degree of swelling. However, further increase in the concentration of comonomers (from 1 to 5 mol/L) leads to the formation of hydrogels, their swelling degree decreases slightly.

As discussed above, the presence of ionogenic groups in the hydrogel structure affects the swelling degree of hydrogels; therefore, the further task of this work is to find the optimal ratio between SPM and HEMA comonomers in terms of the content of ionogenic groups. It turned out that cryogels can be obtained at 80 mol.% SPM relative to HEMA. An increase in the concentration of SPM leads to an increase in the number of SO_3_^−^ groups in the copolymer as confirmed by the ion exchange capacity measurements (Table 1). Due to the dissociation of a large number of -SO_3_K groups (the concentration of which obviously increases with an increase in the SPM content in the reaction system) electrostatic repulsion in the resulting hydrogel between like-charged polymer chains is enhanced, leading to swelling of the polymer network. At the same time, the presence of K^+^ ions in the hydrogel structure leads to an increase in the osmotic pressure inside the gel, which provides additional water sorption by the swelling hydrogel (Figure 2a, red line).

In the P(SPM-HEMA) hydrogel (SPM-HEMA means a copolymer of 3-sulfopropyl methacrylate and hydroxypropyl methacrylate) obtaining the cross-linking is ensured by hydrogen bonds between SPM and HEMA only; therefore, it is impossible to obtain a cryogel at 100% SPM content. Therefore, the SPM hydrogel synthesis was carried out using 0.01 mol.% of *N*,*N*′-methylene-bis-acrylamide (Table 1, experiment 6). However, no gel formation occurs under such conditions, which indicates the predominance of the contribution of hydrogen bonds to the formation of a three-dimensional polymer network. At the same time the introduction of even 0.01 mol.% of MBA to SPM-HEMA reaction system (50/50 monomer ratio, experiment 10) led to a decrease in swelling degree from 435 to 350 g/g (experiments 2 and 10, respectively), which is associated with an increase in the density of cross-links due to additional covalent cross-linking of polymer chains.

The degree of swelling of P(SPM-HEMA) hydrogels also depends on the duration of the synthesis (Figure 2b, black line). The maximum value of degree of swelling is observed when the reaction is carried out for 3 h, while the degree of swelling reaches a plateau when the reaction is carried out for more than 24 h. Meanwhile, the dependence of reaction time on the content of the cross-linked part of the cryogel (gel fraction) has an opposite character. It is seen that when the reaction is carried out for 3 h, the content of the gel fraction is only 65%; as a result, the density of cross-links is quite low and higher values of water sorption are observed. It is worth noting that the content of the sol fraction in the P(SPM-HEMA) hydrogels system decreases as the polymerization reaction proceeds. The sol fraction represents the fraction of the polymer that is not a part of the cross-linked network after a cross-linking reaction occurred [14,36,37,38,39,40]. Thus, after 24 h, the ratio of cross-linked (gel) and non-cross-linked (sol) polymer fractions is 87:13. We believe that although the sol to gel transition occurred within 15 min, complete cross-linking took place up to a few days. At the same time, the sol fraction is absent in the cryogel after 72 h of polymerization; therefore, all the resulting polymer chains are included in the cryogel system.

Taking into account the above mentioned data on hydrogel swelling degree in water, the following conditions for the synthesis of hydrogels are chosen for further research: the total monomer concentration is 1 mol/L; the SPM/HEMA ratio is 50/50 (mol.%); no cross-linker. We chose 24 h as the time of reaction duration since the reaction was characterized by a high yield (90%) under such conditions, and cryogels obtained had moderate values of the degree of swelling, which made it possible to evaluate the effect of additional parameters without completely destroying their structure.

The structure of the gels is confirmed by ATR-FTIR and CP/MAS ^13^C NMR spectroscopies. Figure 3 shows the spectra of sample 2 (Table 1). SPM characteristic bands are observed on the ATR-FTIR spectrum (Figure 3a): 1716 cm^–1^ (v –C=O), 1478 cm^–1^ (v_s_ CH2), 1444 cm^–1^ (v_as_ CH3), 1150 cm^–1^ (v_as_ -C-C(=O)-O + vs. O=S=O), 1038 cm^–1^ (v_s_ S-O), 790 cm^−1^ (v -C-S-O), and 746 cm^–1^ (ρ CH_2_), which corresponds to the literature data [41]. In addition, there is a characteristic band at 1078 cm^–1^ corresponding to –C–OH of the HEMA comonomer, which proves the incorporation of the HEMA comonomer into the gel structure. The chemical shifts characteristic of both SPM and HEMA that are located at 175, 50, 41, and 12 ppm are seen on CP/MAS ^13^C NMR spectrum (Figure 3b), which is consistent with the literature data. In addition, ^13^C NMR-spectrum, similar to ATR-FTIR-spectrum, confirms the presence of the HEMA comonomer in the chemical structure of the gels, as evidenced by the chemical shift at 56 and 63 ppm, corresponding to the carbon atom associated with the HEMA hydroxyl group as well as carbon associated with oxygen.

The macroporous structure of cryogels is studied using scanning electron and optical microscopies. Figure 4 shows the internal morphology of lyophilized samples before and after swelling in water. Before swelling, the gels have interconnected large pores of 50–250 µm in diameter, remaining in place of ice crystals. It should be noted that when increasing the monomer concentration in the reaction system from 1 to 5 wt%, the pore size obviously decreases. In addition, it is shown that the pore structure of cryogel obtained in the presence of 3 wt% DNDs remains unchanged compared to the cryogel obtained under the experiment #2 conditions. (More detailed images of the obtained cryogels are presented in Appendix A). It should be noted that the wall thickness of such a hydrogel does not exceed 10 μm. After equilibrium swelling is established, hydrogels are characterized by interpenetrating pores with a diameter of 150–400 µm. In addition, after swelling, the thickness of the polymer walls decreases. A decrease in the thickness of the polymer walls is also seen in optical microscopy images; it is obvious that during the equilibrium swelling, a part of the unbound polymer chains diffuses into the aqueous dispersion medium and is washed off during its subsequent removal.

### 2.2. Effect of the Additional Agents 

As noted above, natural polymers or other low molecular weight compounds are often introduced into the gel structure to improve the hydrogel swelling degree, as well as other functional properties such as biodegradability, salt resistance, and the potential to be applied as an additional source of microelements. In this work, urea, detonation nanodiamonds (DNDs), starch, and hydroxypropyl cellulose (HPC) are considered as such additives. Starch and HPC are natural polymers and have been used to increase the water-swelling and water retention properties of modified hydrogels. Urea is the main source of nitrogen for crops. The introduction of urea into the structure of the gels can contribute to its slow release, which is necessary to ensure maximum yield.

DNDs are carbon filler, on the surface of which a large number of functional groups are located: hydroxyl, carboxyl, ketone, and lactone ones. The concentration of these groups strongly depends on the purification procedures and subsequent processing of DNDs after detonation synthesis [42]. In [43], DNDs are shown to readily absorb moisture from the environment, with oxidized DNDs showing a stronger interaction with water due to easier hydrogen bonding [34]. In our work, we studied the effect of DND on the process of cryotropic gelation, and also traced the effect of the presence of DND in the cryogel structure on the process of water adsorption and seed germination.

The presence of DNDs in the morphology of cryogels is confirmed by SEM and optical microscopy, where inclusions uniformly distributed over the entire area of the polymer walls are observed. DNDs are known to be unstable and have a tendency towards aggregation. DNDs’ diameter is usually 10–50 nm, but according to our DLS data (Figure 5a), the average size of DNDs used in this work is 185 nm (PDI = 0.28), which indicates the aggregated state of DNDs. Inclusions of similar sizes or slightly larger are observed both in SEM images and optical microscopy photographs. The internal morphology of cryogels with DNDs is similar to the initial cryogels after swelling (Figure 4): there are large, interconnected pores surrounded by polymer walls of 1–3 µm thick. Despite the same wall thickness, polymer walls have a filamentous structure (so the area of polymer walls is smaller), which may be associated with an increase in the degree of swelling of cryogels. ATR-FTIR spectra were recorded for all obtained cryogels (data are given in Appendix A). It is seen that the main characteristic bands corresponding to the initial cryogel are shifted in the presence of urea, DNDs, starch, and HPC, indicating the interaction between components.

It is shown that the addition of urea or DNDs had the greatest effect on the degree of swelling of cryogels (Figure 6a); the cryogel was swollen to equilibrium for a minimum of 24 h. At the same time, the degree of swelling of the gels practically did not depend on the concentration of urea in the system. While a direct dependence of the swelling degree on the DNDs concentration is observed in the range from 0.1 to 3 wt% of DNDs, when using more than 3 wt% of DNDs, no gel forms. The introduction of starch and HPC leads to an increase in the degree of swelling only at low concentrations (no more than 0.5 wt%). A further increase in the content of dopants led to a sharp decrease in the degree of swelling. Starch and HPC are natural polymers that tend to gel. Apparently, at additive content of more than 0.5 wt%, the formation of a second interpenetrating network begins, which works by analogy with an increase in the density of cross-links and, as a result, the degree of swelling of hydrogels in water decreases.

Water, despite being an important component for the growth of living organisms, is still an ideal solvent. In practice, solutions containing various micro- and macroelements are always used. In this regard, the swelling degree of cryogels in NaCl solutions (0.9%), as well as in the Knop’s solution, was studied. Knop’s solution is a widely used nutrient medium for plants and presents a mixture of Ca(NO_3_)_2_ ∙ 4H_2_O (1 g/L), KNO_3_ (0.25 g/L), KCl (0.12 g/L), KH_2_PO_4_ (0.25 g/L), MgSO_4_ ∙ 7H_2_O (0.25 g/L) and micro elements (iron, boron, manganese, zinc, copper). Cryogels obtained in the presence of 3 wt% dopant are chosen for the study, since the maximum effect of its introduction into the cryogel structure is observed at this concentration. The replacement of water with saline solutions led to a sharp decrease in the cryogels swelling degree (Figure 6b). The swelling of cryogels in Knop’s solution is 70 *w*/*w* for the initial cryogel, and 47 *w*/*w* for cryogels prepared in the presence of HPC and DNDs. A decrease in swelling degree by more than 30% indicates a significant contribution of ion–ion interactions to the swelling process. 

The ability to retain moisture is the main property of hydrogels used for agriculture. The water-retaining properties are studied for cryogels obtained in the presence of 3 wt% of dopant also (Figure 6c). The use of starch and HPC makes it possible to slow down the drying rate of gels by 13% and 3.5%, respectively (relative to the initial cryogel without the use of agents). While for cryogels with DNDs and urea, the water-retaining properties deteriorate by 4.5 and 2%, respectively. This may be due to large pores, from which the solvent (that fills the space of these pores) evaporates faster.

The cryogels obtained present systems with a soft structure. Since no cross-linking agent is added into the system under study and the maintenance of the hydrogel structure is ensured by non-covalent interactions only, the additives make an additional contribution to the softening of the hydrogel structure due to destroying a part of the intramolecular interactions of the cryogel and the inclusion of additives (see the photos in Appendix A). By this reason, it becomes impossible to evaluate a compressive strength of hydrogels with additives. It is worth noting that the inclusion of additives into the cryogel was aimed to optimize water swelling of cryogel by modification of its composition rather than to obtain a more stable cryogel. Nevertheless, we were successful in measuring the compressive strength of the initial cryogel (sample 2).

The dependence of compressive stress on the strain before the swelling is presented below (Figure 7). The estimated Young’s modulus of initial cryogel before the swelling is equal to 1.6 MPa. The study the mechanical properties of cryogels after the swelling and removal of reaction products turned out to be impossible, since the obtained cryogels did not maintain a stiff shape and had a loose structure.

### 2.3. Influence of the Gel on the Germination of Watercress Seeds

In order to investigate the ability of hydrogels to deliver nutrients (micro- and macroelements) and water to the root system of plants, the studies on the cultivation of *Lepidium sativum* L. variety Ajur using hydrogels as a substrate are carried out. The following samples are involved in this investigation: without an additional agent (total monomer concentration is 1 mol/L), and with 3 wt% of urea, starch, HPC, and DNDs (Table 2). A sample without cryogel is used as a control. 

Judging by the reaction of the phyto-test object, *Lepidium sativum* L. variety Ajur, at the early stages of its development, the presence of hydrogel around its seeds does not significantly affect their germination (germination energy, germinability) and morphometric characteristics of seedling growth. There is a slight tendency to stimulate the germination energy of *Lepidium sativum* L. variety Ajur seeds on the third day, which slightly increases on the seventh day in terms of seed germinability in the hydrogel obtained in the presence of HPC or DNDs (Table 2). For hydrogels obtained in the presence of starch or without additives, seed germinability is also stimulated in the form of a weak unreliable trend. For a hydrogel synthesized in the presence of urea, the values of germination energy and seed germinability do not differ from those in the control. Figure 8 shows typical photographs of seed germination over time on a cryogel obtained in the presence of DNDs.

Changes in morphometric characteristics of *Lepidium sativum* L. variety Ajur are more pronounced in sprouts than in roots. Thus, for all experiments, an increase in the form of a trend in the length of sprouts (by 4–13%) is observed, while the length of the roots slightly increases (by 2–9%) only for hydrogels without additives and with additives such as urea and hydroxypropyl cellulose (Table 2, samples no. 1, 2, and 4).

However, the noted trends in the change in the morphometric characteristics of sprouts turn into reliable ones in relation to their weight parameters (Table 3). So, in all hydrogels, with the exception of the gel with the addition of starch (Table 3, samples No. 1, 2, 4, 5), the raw weight of the sprouts is higher than the control one by 12–34%. The dry weight of the sprouts is significantly lower (by 20%) than the control in the variants with hydrogels obtained in the presence of urea or without additives. At the same time, the dry weight of the sprouts is unreliably lower (by 7%) for hydrogels obtained in the presence of starch and DNDs. For the hydrogel obtained in the presence of HPC, the dry weight of the sprouts does not differ from that in the control sample. The dry matter in sprouts is significantly reduced in almost all experiment variants by 24–37%, with the exception of the hydrogel containing starch.

Comparison of the data obtained reveals an interesting picture of the influence of hydrogels, predominantly stimulating morphometric characteristics and wet biomass and reducing the values in relation to dry weight and dry matter of *Lepidium sativum* L. variety Ajur sprouts. Thus, it is found that the correlation coefficient between the values of the wet weight of sprouts by variants and the values of dry matter content in them is R = −0.86, between the values of the dry weight of sprouts and the dry matter content in them, R = 0.63, and between the values wet and dry mass of sprouts, R = −0.18. The noted high negative correlation between the fresh weight of the sprouts and their dry matter content, a positive correlation between the dry weight of the sprouts and their dry matter content, and the absence of a significant correlation between wet and dry weights in the experimental variants with a weakly stimulating change in the morphometric characteristics of seedlings in relation to control, together indicate a high biological activity of hydrogels, whose presence around sprouting seeds obviously contributes to the activation of the metabolism in plants at the early stages of their development and the intensification of growth processes with a regular decrease in nutrients of an organic and mineral nature in plants, which is reflected accordingly in the values of dry matter and dry mass.

Thus, the features of seed germination of the phyto-test object of *Lepidium sativum* L. variety Ajur in a thin layer of hydrogels of various compositions indicate the high biological activity of most of them, which tend to have a stimulating effect on seed germination and the morphometric characteristics of seedlings and contribute to a significant increase in the wet mass of seedlings with a decrease in the dry mass of the latter and the content of dry matter in them. The noted effects are least pronounced for the hydrogel with the addition of starch, where the plant parameters taken into account did not differ from those in the control. Analyzing the obtained data in aggregate and taking into account changes in seed germination rates, the morphometric characteristics of seedlings, their wet and dry weight, and their dry matter content—the most promising in terms of impact on the physiological state and growth rates of plants in the early stages of their development—are hydrogels with HPC and DND additives. Based on the results of the present study, we may make the following assumption on factors that may contribute to the beneficial influence hydrogels with HPC and DNDs. Firstly, it is the influence of cryogels on the first stage of seed germination that includes the process of water uptake by the mature seed called imbibition [44]. As seen from Figure 6c, during first 2–15 h, water release is more intensive for the initial cryogel and that with HPC and DNDs additives. Secondly, it is known that when seed germination occurs in the presence of a gel, the gel adheres easily to the seed, forming the protective coating [45]. Despite the cryogels obtained presenting a soft structure, the measurement of their mechanical properties turned to be impossible; the appearance of the cryogel (see photo in Appendix A) shows that hydrogels with HPC and DNDs are characterized by less dense structures, facilitating distribution in the substrate and around seeds.

It is of great interest in the future to study the mechanisms of the influence of all variants of the tested hydrogels on plants throughout their ontogenesis and to evaluate their effectiveness according to integral indicators of the optimal cultivation of plants, i.e., the yield and quality of the resulting plant products.

## 3. Materials and Methods

### 3.1. Materials 

Potassium 3-sulfopropyl methacrylate (SPM), *N*,*N*′-methylenebis(acrylamide) (MBA), *N*,*N*,*N*′,*N*′-tetramethylethylenediamine (TEMED) were purchased from Sigma-Aldrich (Darmstadt, Germany) and used without preliminary purification. 2-Hydroxyethyl methacrylate (HEMA) (Sigma-Aldrich, Darmstadt, Germany) was purified by dint of molecular sieve. Potassium persulfate (K_2_S_2_O_8_), HCl, NaCl, and NaOH were purchased from Vekton LLC (Saint-Petersburg, Russia) and purified by recrystallization from ethanol. Hydroxypropyl cellulose (HPC) (Across Organic, Geel, Belgium), urea (Vekton LLC, Saint-Petersburg, Russia), denotational nanodiamonds (DNDs) (FGYP SKTB ‘Technolog’, Saint-Petersburg, Russia), and starch (Vekton LLC, Saint-Petersburg, Russia) were used as received. Knop’s Solution: Ca(NO_3_)_2_ × 4H_2_O (1 g/L), KNO_3_ (0.25 g/L), KCl (0.12 g/L), KH_2_PO_4_ (0.25 g/L), MgSO_4_ × 7H_2_O (0.25 g/L) and microelements (iron, boron, manganese, zinc, copper). Knop’s Solution (Vekton LLC, Saint-Petersburg, Russia) was used as received.

Double distilled water was used to prepare solutions and carry out the polymerization. Water specific conductivity (0.1 S/cm) and the surface tension (72 mN/m) proved that it was free of surface-active impurities.

### 3.2. Preparation of Cryogels

The syntheses of cryogels were carried out by dissolving of appropriate amounts of SPM and HEMA as well as an initiating system in 1 mL of water. The polymerization was initiated by the red-ox initial system (K_2_S_2_O_8_ 0.8 mol.% to monomers and TEMED 0.1 mol.% to monomers). The initial reaction solution was prepared in plastic molds using a cooling bath (the temperature range was 2–5 °C) and kept in a freezer at −18 °C for 24 h. Afterwards, the plastic molds were kept at room temperature until the complete thawing of the reaction system. The result cryogels were kept in excess water at least for 2 days to remove the unbounded reaction products. Finally, the cryogels were freeze-dried, before further use. Concentrations of monomers are given in Table 1. 

When additives such as urea, HPC, and DNDs were used, the same procedure of cryogel synthesis was applied. The concentrations of the agents were varied in a range of 0.1–7.5 wt% (to monomers)**.** In case of using starch as an additive, it was dissolved in a hot water on the first stage of polymerization, then the solution was cooled down and the procedure was followed as described above. 

### 3.3. Characterizations of Cryogels 

#### Scanning Electron and Optic Microscopies

SEM images of cryogels were obtained using field emission scanning electron microscope Zeiss SUPRA 55 VP (Oberkochen, Germany). The pore structure of the obtained cryogels was studied in the swollen state. For this purpose, gel samples were exposed to water until the equilibrium was established, then frozen rapidly and lyophilized. Freeze-dried pieces of cryogels were placed on a silicon wafer with next sputtering with 10 nm of AuPd. Afterwards, the samples were inserted into the observation electron microscope chamber.

The optical microscopy photographs were obtained using optic microscope MIKMED-5 (Saint-Petersburg, Russia).

### 3.4. Dynamic Light Scattering 

Sizes and polydispersity index (PDI) of DNDs were measured in their dilute solution using a Zetasizer Nano ZS instrument (Malvern Panalytical, Malvern, UK) at 25 °C. The instrument was operating at a wavelength of 633 nm and the measurements were carried out at a detection angle of 90°. For each sample, at least three measurements were taken to check the repeatability of the results and the analysis was conducted using CONTIN analysis mode.

### 3.5. ATR-FTIR and ^13^C NMR Spectroscopies

Since the samples are insoluble, spectral studies were carried out for freeze-dried powders. The ATR-FTIR spectra of the cryogels were recorded in the range of 500–2000 cm^−1^ wavenumbers using an IR-Affinity-1S spectrometer (Shimadzu, Kyoto, Japan). All the spectra represent an average of 32 scans taken.

The structure of the cryogels was investigated by solid state CP/MAS ^13^C NMR spectroscopy (Cross-Polarization Magic-Angle-Spinning NMR). The NMR spectra were recorded on an AVANCE II-500 WB NMR spectrometer (Bruker, Billerica, MA, USA) operating at resonance frequency of 125.8 MHz (CP/MAS ^13^C NMR). Polymer samples were packed into zirconium rotors with a diameter of 4 mm, the spectra were registered at a temperature of 20 °C and rotation frequency of 10 or 13 kHz.

### 3.6. Swelling Tests in Different Solutions

The swelling degree (SW) of each sample was measured using a gravimetric method. Swelling tests were conducted under identical conditions for all the cryogels. The solvent volume was 600 mL, and the gel size was 8 mm in diameter and 15 mm in height. The cryogel was swollen to equilibrium for a minimum of 24 h. After swelling equilibrium was reached (24 h), the hydrogels were removed from the swelling medium, the excess surface water was lightly surface dried with filter paper, and the sample was weighed. Swelling tests were carried out in bi-distilled water, NaCl solution (0.9%), and Knop’s solution. Firstly, the dry samples were weighted (Md); afterwards, the samples were immersed into excessive water until equilibrium swelling. Excess liquid was removed from the swollen cryogels surface and weighted as Msw. The swelling ratio was calculated by the following Equation (1):(1)$$\mathrm{SW}=\frac{M_{\mathrm{sw}}-M_{d}}{M_{d}},$$
where Md and Msw is the weight of the sample in the dry and swollen state, respectively.

### 3.7. Water Retention

Dried samples (0.05 g) were maintained in 10 mL of double distilled water for swelling equilibrium at a room temperature. The swollen sample was weighed (Msw) and then left for drying in a room temperature and weighted at certain intervals (M_t_). The water retention ratio at different times was calculated by Equation (2).
(2)$$\mathrm{WR}=1-\frac{M_{\mathrm{sw}}-M_{t}}{M_{\mathrm{sw}}}\cdot 100\%,$$

### 3.8. Ion Exchange Capacity

The ion exchange capacity (IEC) of the synthesized cryogels was measured by acid-base titration. Typically, 0.01 g of the sample was treated with 10 mL of 0.5 M HCl for 1 day. The sample was then purified in excess of water to remove any excess of acid and salt. After freeze-drying, the sample was treated with 20 mL of 20% NaCl solution for 6 h to exchange all H^+^ ions with Na^+^. The HCl formed as a result of this ion exchange was titrated with 0.01 M NaOH, using phenolphthalein as the indicator. The ion exchange capacity was calculated using Equation (3).
(3)$$\mathrm{IEC}=C_{T}\frac{V_{T}}{m_{\mathrm{gel}}},$$
where C_T_ is titrant concentration, V_T_ is titrant volume, and m_gel_ is the weight of the gel.

### 3.9. Measurement of Mechanical Properties of Cryogel by Dynamic Mechanical Analyzer (DMA)

The compression test was conducted for the cryogel immediately after synthesis (before swelling). Compression tests were performed using a tension-compression test machine (DMA INSTRON 5943, Waltham, MA, USA) using samples of cylindrical shape (~10 mm diameter and ~70 mm height). Samples were compressed with two parallel plates at the maximum loading of 0.01 N with a compression rate of 1 mm/min. The stress, strain (%), and toughness values were calculated by the Trapezium X Materials Testing Software (Shimadzu, Japan). The compressive modulus was calculated from the linear region of the stress-strain curve (0–40% strain).

### 3.10. Biological Activity

The biological activity of the synthesized hydrogels was determined in laboratory conditions at an air temperature of +22 to +24 °C and an air humidity of 60 ± 5% to accommodate seed germination and seedling growth of a phyto-test culture—*Lepidium sativum* L. variety Ajur. Plant seeds were obtained from the collection of the NI Vavilov Institute of Plant Genetic Resources (VIR). The seeds were germinated for 7 days in Petri dishes on an inert hydrophilic material, which ensures a uniform supply of periodically introduced distilled water (on the 1st, 3rd, 5th–7th days, 5 mL/Petri dish) to the seedling roots. The variant without hydrogel served as the control; in the experimental variants, the hydrogel suspension was applied in a thin layer (1 mm thick) on the surface of the hydrophilic material, after which the seeds were distributed over the area of the material with the gel.

The preparation of working suspensions of hydrogels applied to the material was carried out by diluting them with distilled water in a ratio of 1:500 (previously experimentally selected as the most optimal for seed sprouting).

The study was carried out in accordance with the rules of the international association (ISTA) and standards (GOST 12038-84). On the 3rd day after sowing the seeds, the emergence rate was determined, on the 7th day after sowing the seeds, the germination of seeds was evaluated, and the length of the sprouts and roots, the biomass of the sprouts, and their dry matter content were also measured. The experiment was repeated three times, the number of seeds per variant was 400 pieces.

### 3.11. Statistical Analysis

The experiments were performed with n = 3–4. All data measurements were represented as average ± standard deviations.

## 4. Conclusions

In this work, physically cross-linked biocompatible cryogels based on a copolymer of potassium 3-sulfopropyl methacrylate and 2-hydroxyethyl methacrylate were obtained using potassium persulfate as an initiator. The use of the cryotropic gelation method made it possible to obtain porous polyelectrolyte gels at a minimum concentration of comonomers in the reaction system. It has been shown that the optimization of the conditions for the synthesis of cryogels made it possible to increase the degree of their swelling to 1070 g/g, which is three times higher than the degree of swelling of the hydrogels based on natural polymers that are most commonly used in agriculture. It has been shown that cryogels obtained in the presence of nanodiamonds exhibit high biological activity during the germination of *Lepidium sativum* L. variety Ajur seeds, which manifests itself by stimulating seed germination and a significant increase in the raw weight of sprouts. In addition, cryogels containing HPC and DNDs can be considered the most promising in terms of their effect on the physiological state and growth rates of plants at the early stages of their development. Thus, the synthesized cryogels are of interest for use as independent substrates (without the use of soil) for the delivery of water and nutrients in the process of growing plants in a zone of unstable moisture.

One of the promising directions for further research of the obtained cryogels that is worth mentioning is the preparation of interpolymer complexes using the cryogel as a template. For example, the formation of an interpolymer complex based on sulfo-containing cryogels and electrically conductive polymers is of practical interest in the development of biosensors.

## Figures and Tables

**Figure 1 ijms-24-02949-f001:**
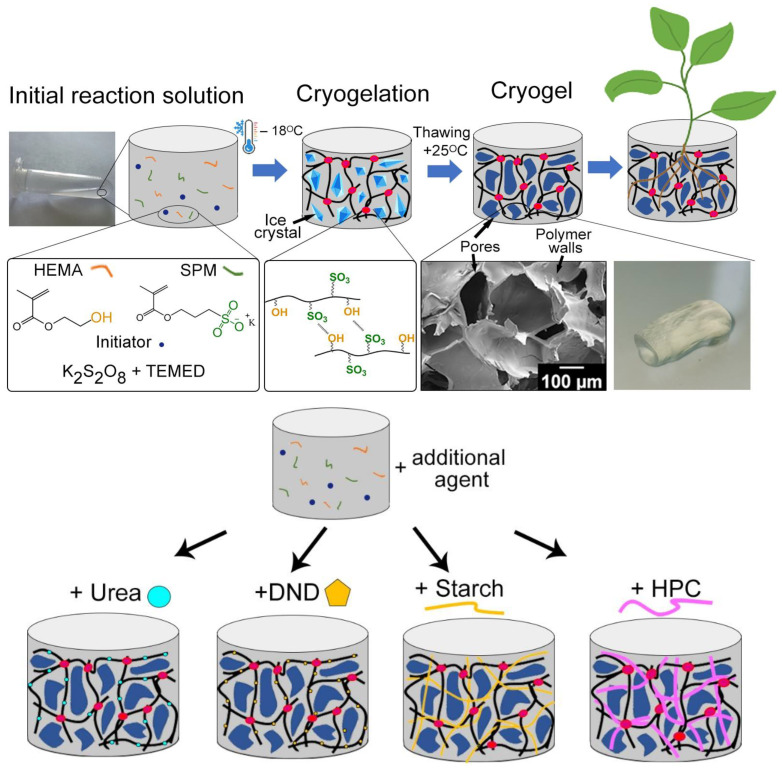
Scheme of cryogel synthesis and cryogel appearance.

**Figure 2 ijms-24-02949-f002:**
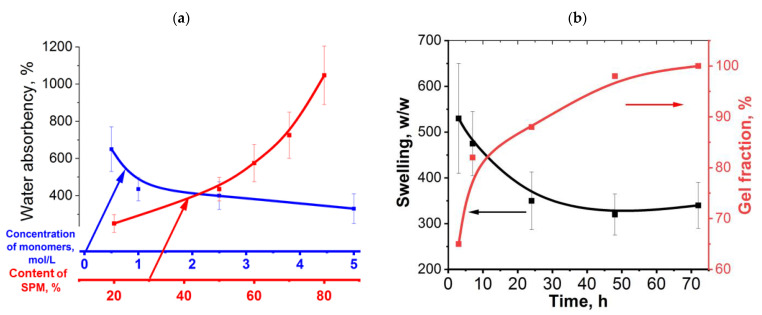
Dependences of the water absorbency/swelling degree of cryogels in distilled water on the total concentration of monomers ((**a**), blue line), on SPM content in the system ((**a**), red line), on the reaction time ((**b**), black line) and dependence of the content of the gel fraction in the system vs. the reaction time ((**b**), red line). The arrows indicate the corresponding dependences shown in color.

**Figure 3 ijms-24-02949-f003:**
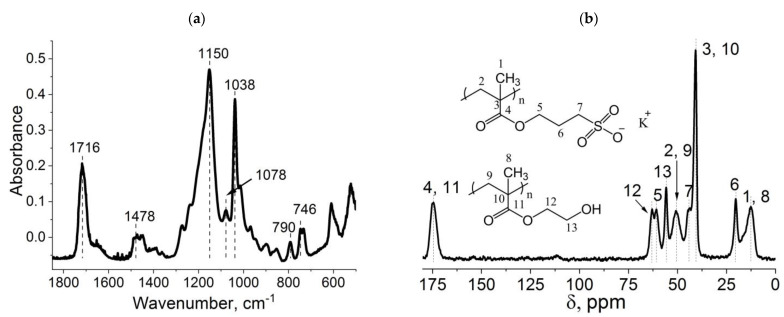
ATR-FTIR (**a**) and CP/MAS ^13^C NMR (**b**) spectra of obtained cryogels (sample 2).

**Figure 4 ijms-24-02949-f004:**
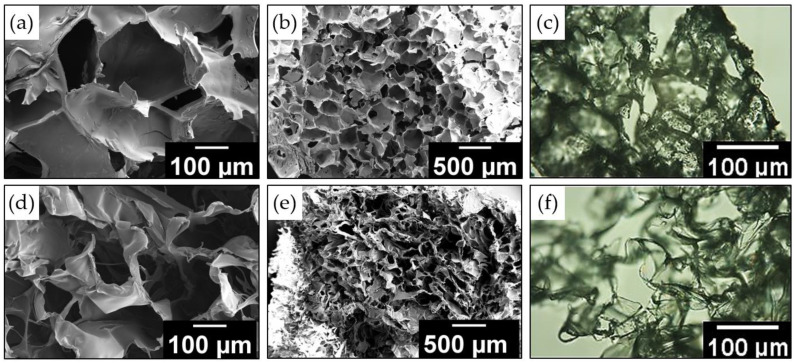
SEM (**a**,**b**,**d**,**e**) and optical microscopy (**c**,**f**) images of the cryogel sample 2 before (**a**–**c**) and after swelling (**d**–**f**).

**Figure 5 ijms-24-02949-f005:**
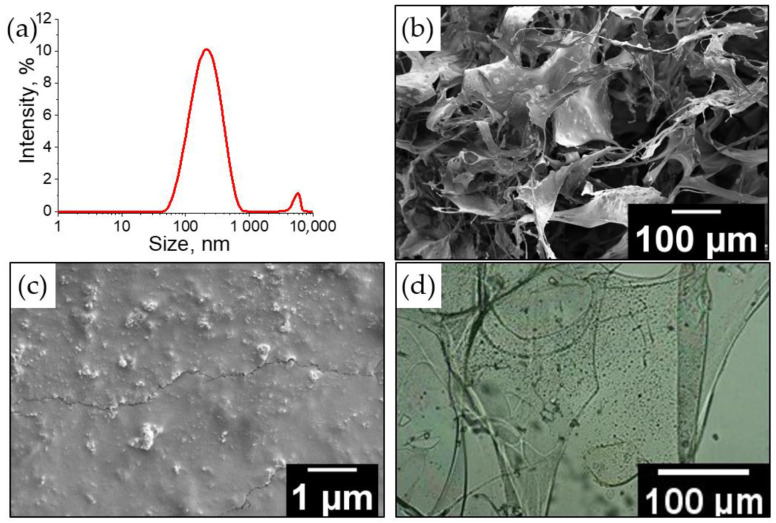
DLS for DNDs (**a**); SEM (**b**,**c**) and optic microscopy (**d**) images of cryogels obtained in the presence of 3 wt% of DNDs.

**Figure 6 ijms-24-02949-f006:**
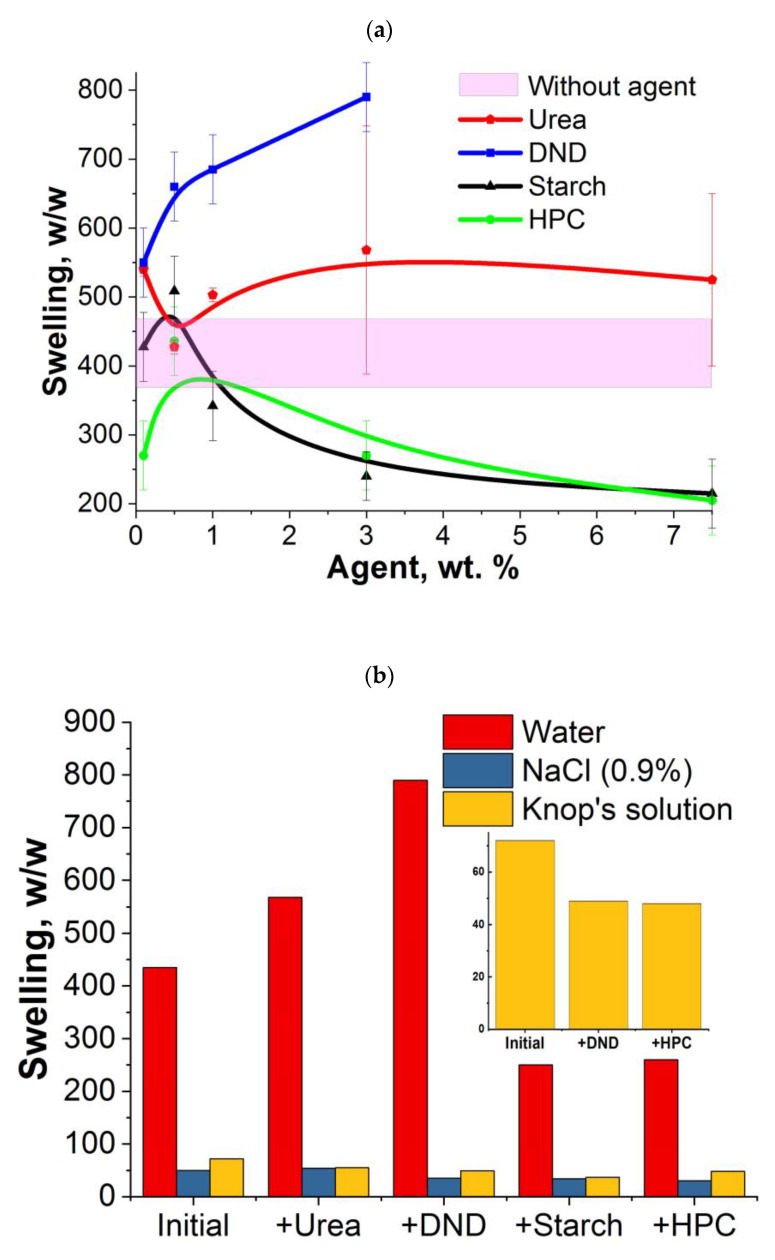
Dependences of the swelling degree of cryogels vs. content of additional agents (**a**); sorption properties of cryogels (obtained in the presence of 3 wt% of additional agents) in various solutions (**b**); water retention of cryogels (**c**).

**Figure 7 ijms-24-02949-f007:**
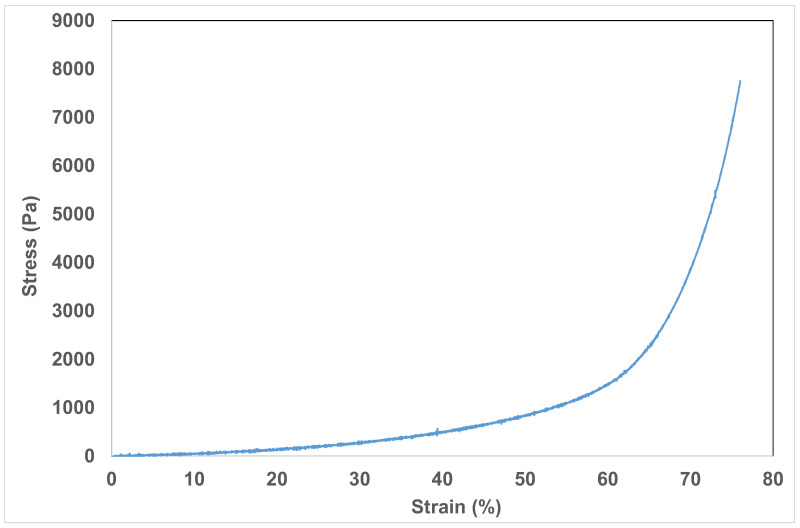
The compression stress-strain curve of cryogel (sample 2).

**Figure 8 ijms-24-02949-f008:**
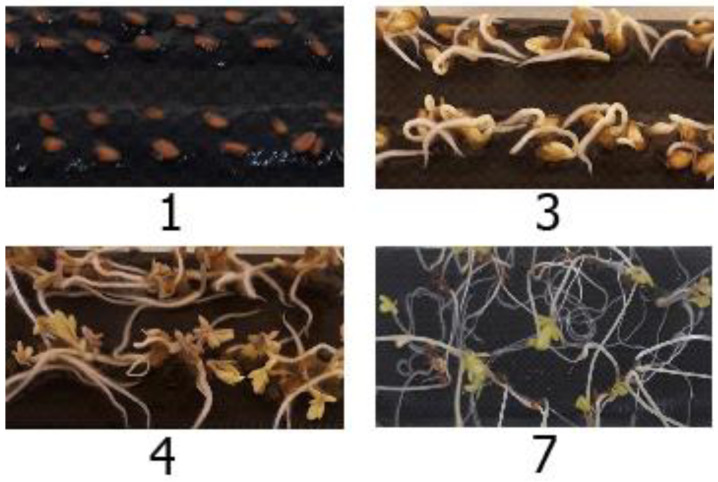
Typical photographs of seed germination stages (1st, 3rd, 4th, and 7th days) (for gel prepared in the presence of DNDs).

**Table 1 ijms-24-02949-t001:** Composition of the reaction system for the synthesis of P(SPM−HEMA) cryogels and cryogel characteristics.

Sample	SPM/HEMA, mol.%	∑C_monomer_, mol/L	SO_3_, 10^−3^ mol−eq/g	Swelling Degree, g/g
Effect of temperature				
1 *	50/50	1	No gel formation	
2	50/50	1	2.61 ± 0.17	435 ± 35
Effect of monomer concentration				
3	50/50	0.5	2.67 ± 0.18	650 ± 45
2	50/50	1	2.61 ± 0.16	435 ± 32
4	50/50	2.5	2.76 ± 0.21	400 ± 36
5	50/50	5	2.66 ± 0.17	320 ± 22
Effect of monomer ratio				
6 **	100/0	1	No gel formation	
7	80/20	1	3.05 ± 0.22	1070 ± 85
8	70/30	1	3.53 ± 0.23	725 ± 50
9	60/40	1	2.73 ± 0.19	575 ± 46
2	50/50	1	2.61 ± 0.15	435 ± 30
10 **	50/50	1	2.67 ± 0.17	350 ± 24
11	20/80	1	1.21 ± 0.11	250 ± 17

* The temperature was +25 °C. ** MBA = 0.01 wt% to monomers

**Table 2 ijms-24-02949-t002:** Influence of hydrogel on sowing characteristics and growth rates of *Lepidium sativum* L. variety Ajur seedlings.

Sample No.	Agent, (3 wt%)	Emergence rate		Germination		Sprout Length		Root Length	
		%	% to Control	%	% to Control	cm	% to Control	cm	% to Control
1	-	88	101	96	103	5.1 ± 0.3	111	4.5 ± 0.4	102
2	Urea	87	100	93	100	5.1 ± 0.3	111	4.8 ± 0.5	109
3	Starch	87	100	97	104	4.8 ± 0.4	104	4.2 ± 0.5	95
4	HPC	91	105	100	108	5.2 ± 0.5	113	4.7 ± 0.5	107
5	DNDs	90	104	99	106	5.1 ± 0.4	111	4.4 ± 0.4	100
6	(control sample) without cryogel	87	100	93	100	4.6 ± 0.4	100	4.4 ± 0.4	100

**Table 3 ijms-24-02949-t003:** Influence of adding hydrogel to the root environment on the accumulation of raw, dry weight and dry matter by *Lepidium sativum* L. variety Ajur seedlings.

Sample No.	Agent, (3 wt%)	Wet Weight 100 Sprouts		Dry Weight 100 Sprouts		Dry Matter of Sprouts	
		g	% to Control	g	% to Control	%	% to Control
1	-	2.48	133 *	0.12	80 *	5.0	63 *
2	Urea	2.08	112 *	0.12	80 *	5.7	71 *
3	Starch	1.82	98	0.14	93	7.9	99
4	HPC	2.46	132 *	0.15	100	6.1	76 *
5	DNDs	2.50	134 *	0.14	93	5.6	70 *
6	(control sample)	1.86	100	0.15	100	8.0	100

* the value significantly differs from the control at the 5% significance level.

## Data Availability

The authors declare that all data supporting the findings are available within the paper or are available from the authors upon request.

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
