# Peer review of "Sulfonic Cryogels as Innovative Materials for Biotechnological Applications: Synthesis, Modification, and Biological Activity"

_ijms, 2023, doi:10.3390/ijms24032949_

Round 1

Reviewer 1 Report

 The idea is interesting and the combination of the components chosen seems to be original. However, the materials characterization is not complete, particularly regarding the cryogels that contain HPC. The presence of HPC or DNDs was beneficial for the plant germination, but a solid discussion on this is missing.

All data require statistical analyses and error bars. The lack of standard deviations and statistical analyses derailed a critical analysis on the results.

Fig. 3a – “y” axis, dot instead of comma

L233 – optical microscopy

L240-243 -  The determination of the wall thickness of the cryogels by SEM images is inaccurate. Please remove the quantitative analyses on the wall thickness. The same on L273.

L276 – Please indicate after how many hours the swelling degree was determined.

L306-308 – “At the same time, the presence of a second interpenetrating HPC network in the structure of the gels made it possible to reduce the difference in swelling from 83% (for the initial gel) to 81%. “ Here I see two problems, the first is that the second interpenetrating HPC network was not proved by any experimental data. The second is that 2% might be within the error and probably cannot be considered as a difference.

L329-330 – “For hydrogels obtained in the presence of starch  or without additives, seed germinatability is also stimulated in the form of a weak unreliable trend”, what exactly do the authors mean by “unreliable trend”? If it is not reliable, the results should be removed from the manuscript.

L507 – “…cryohydrogels containing HPC and DNDs can be considered as the most promising…” why?

Reviewer 2 Report

The author should revise the manuscript as following suggested points

1.  Add the chemical structures and schematic illustration of the hydrogel chemistry and crosslinking along with it add the picture of the hydrogel in figure no. 1 

2. Explain the sol fraction value (non-gel fraction)  for it authors should follow the following articles. A. Self-assembly of partially alkylated dextran-graft-poly [(2-dimethylamino) ethyl methacrylate] copolymer facilitating hydrophobic/hydrophilic drug delivery and improving conetwork hydrogel properties. Biomacromolecules19(4), 1142-1153.

3. The author should evaluate the hydrogels' degradation profile and describe it in the discussion part of the manuscript.

4. In Figure 2B, Fig 6 A and B the Y-axis author should correct by Swelling w/w instead of SW g/g

5. Author must add the different stage of  germination Photographs in the manuscript

6.AUthor should add the compressive strhength of the hydrogels ( before swelling and adter swelling) in the revised manuscript.

7. Author should cite the following hydrogels article in the manuscript A.Dually crosslinked injectable hydrogels of poly (ethylene glycol) and poly [(2-dimethylamino) ethyl methacrylate]-b-poly (N-isopropyl acrylamide) as a wound healing promoter. Journal of Materials Chemistry B5(25), 4955-4965. B.Liquid prepolymer-based in situ formation of degradable poly (ethylene glycol)-linked-poly (caprolactone)-linked-poly (2-dimethylaminoethyl) methacrylate amphiphilic conetwork gels showing polarity driven gelation and bioadhesion. ACS Applied Bio Materials1(5), 1606-1619. C.Ultrahigh water-retention cellulose hydrogels as soil amendments for early seed germination under harsh conditions. Journal of Cleaner Production370, 133602. D.A Systematic Review of the Potential of a Dynamic Hydrogel as a Substrate for Sustainable Agriculture. Horticulturae8(11), 1026. ESuperabsorbent Polymer Hydrogels for Sustainable Agriculture: A Review." Horticulturae 8, no. 7 (2022): 605.

8. Athor should revise the all graphs and figure make them more clear and unified size and shape 

Reviewer 3 Report

In this study, the authors report the synthesis and characterization of potassium 3-sulfopropyl methacrylate and 2-hydroxyethyl methacrylate cryogels with applications in agriculture. The manuscript is well-written, and the experimental results support the statements. I suggest only minor revisions before publication:

1.       Please perform and English review and check the manuscript for typos.

2.       Some of the figures and tables are placed in the middle of paragraphs making the text hard to read, please review (e.g. lines 151-159)

3.       Are there any previous literature reports regarding the use of DND, HPC, urea, or starch as additives for these types of cryogels? A paragraph regarding these studies could be inserted in the introduction with appropriate literature references.

4.       Are there any other fields where the applications of these materials could be extended? The authors could mention these in the conclusions and future trends section.

Round 2

Reviewer 1 Report

I think the authors revised the manuscript in a proper way. All my questions were addressed and I think the manuscript can be accepted.

Reviewer 2 Report

The Authors have revised all suggested points